# Protocol for an international multicenter randomized controlled trial assessing treatment success and safety of peroral endoscopic myotomy vs endoscopic balloon dilation for the treatment of achalasia in children

Carlijn Mussies[ID][1][☯], Marinde van Lennep[1][☯], Johanna H. van der Lee[2,3], Maartje J. Singendonk[1], Marc. A. Benninga[1], Barbara A. Bastiaansen[4], Paul Fockens[4,5], Albert J. Bredenoord[4], Michiel P. van Wijk[ID][6]*

1 Emma Children's Hospital—Amsterdam UMC, Location University of Amsterdam, Pediatric Gastroenterology, Amsterdam, The Netherlands, 2 Emma Children's Hospital—Amsterdam UMC, Location University of Amsterdam, Pediatric Clinical Research Office, Amsterdam, The Netherlands, 3 Knowledge Institute of the Dutch Association of Medical Specialists, Utrecht, The Netherlands, 4 Gastroenterology & Hepatology, Amsterdam UMC, Location University of Amsterdam, Amsterdam, The Netherlands, 5 Gastroenterology & Hepatology, Amsterdam UMC, Location Vrije Universiteit Amsterdam, Amsterdam, The Netherlands, 6 Emma Children's Hospital—Amsterdam UMC, Location Vrije Universiteit Amsterdam, Amsterdam, The Netherlands

☯ These authors contributed equally to this work.
* m.vanwijk@amsterdamumc.nl

## Abstract

### Introduction

Achalasia is a rare neurodegenerative esophageal motility disorder characterized by incomplete lower esophageal sphincter (LES) relaxation, increased LES tone and absence of esophageal peristalsis. Achalasia requires invasive treatment in all patients. Conventional treatment options include endoscopic balloon dilation (EBD) and laparoscopic Heller's myotomy (LHM). Recently, a less invasive endoscopic therapy has been developed; Peroral Endoscopic Myotomy (POEM). POEM integrates the theoretical advantages of both EBD and LHM (no skin incisions, less pain, short hospital stay, less blood loss and a durable myotomy). Our aim is to compare efficacy and safety of POEM vs. EBD as primary treatment for achalasia in children.

### Methods and analysis

This multi-center, and center-stratified block-randomized controlled trial will assess safety and efficacy of POEM vs EBD. Primary outcome measure is the need for retreatment due to treatment failure (i.e. persisting symptoms (Eckardt score > 3) *with* evidence of recurrence on barium swallow and/*or* HRM within 12 months follow-up) as assed by a blinded end-point committee (PROBE design).

**Data Availability Statement:** No datasets were generated or analysed during the current study. All relevant data from this study will be made available upon study completion.

**Funding:** This study is funded by the For Wishdom Foundation (https://www.forwishdom.org/). The funders had no role in study design, data collection and analysis, decision to publish, or preparation of the manuscript.

**Competing interests:** The authors have declared that no competing interests exist.

## Discussion

This RCT will be the first one to evaluate which endoscopic therapy is most effective and safe for treatment of naïve pediatric patients with achalasia.

## Introduction

Achalasia is a neurodegenerative esophageal motility disorder characterized by incomplete lower esophageal sphincter (LES) relaxation, increased LES tone and absence of esophageal peristalsis. Symptoms include dysphagia, regurgitation, chest pain and weight loss [1]. In children, achalasia is mostly diagnosed after the age of 7 years and has an estimated incidence of 0.1–0.18/100.000 children per year.]

If left untreated, patients are likely to become tube-feeding dependent. Additionally, dilation of the distal esophagus ("megaesophagus") may occur.

When symptoms suggestive of achalasia are present, a timed barium esophagram can be performed. Signs suggestive of achalasia include esophageal stasis, esophageal dilation above the LES and tapering of the LES (the so called "bird's beak"). However, a barium esophagram may be normal despite presence of the disease [2].

The gold standard test for diagnosing achalasia in both children and adults, is esophageal high resolution manometry (HRM) [2]. According to the Chicago Classification of esophageal motility disorders (CC), achalasia is present when the integrated LES relaxation pressure (IRP4, measured with HRM) is above the upper limit of normal, combined with absent esophageal peristalsis or esophageal spasm (see Fig 1) [3]). The CC is able to detect three different subtypes of achalasia (type 1: aperistalsis with absence of pressurization; type 2: aperistalsis with panesophageal pressurization and type 3: spastic) [4].

### Current treatment options

Treatments of achalasia are aimed at reducing LES pressure. This leads to improvement of esophageal emptying, thereby decreasing symptoms and prevention of a megaesophagus in the long-term. In children, the optimal therapy of choice is still under debate [5, 6].

**Endoscopic balloon dilation (EBD).** EBD is considered the most effective *non-surgical* treatment for achalasia in adults and is offered as first-line treatment in children in most pediatric achalasia centers [1]. However, many children with achalasia are in need of multiple relapse dilations after initial EBD and a substantial group eventually requires surgery [5]. Adult literature has suggested that younger patients (men <40 years) have a shorter clinical response to EBD [3]. A review including only pediatrics studies, comparing EBD with laparoscopic Heller's myotomy (LHM) found relapse rates to range from 71% to 90% after EBD compared with 22% to 39% after LHM (follow-up 1–22 years).[1, 7–9] In these pediatric studies, patients mostly received one single dilation. However, it is known that success rates in EBD increase if a second dilation session as part of the initial therapy is scheduled few weeks apart from the initial dilation [10–13].

**Laparoscopic Heller's myotomy (LHM).** LHM is a surgical alternative for EBD. During LHM, the circular muscle fibers of the LES are cut. In most pediatric achalasia centers, LHM is offered as a second-step therapy after EBD therapy has failed [14]. Compared with EBD, LHM is a more invasive technique as it requires abdominal laparoscopic surgery and a prolonged hospital stay [1, 7, 8, 15].

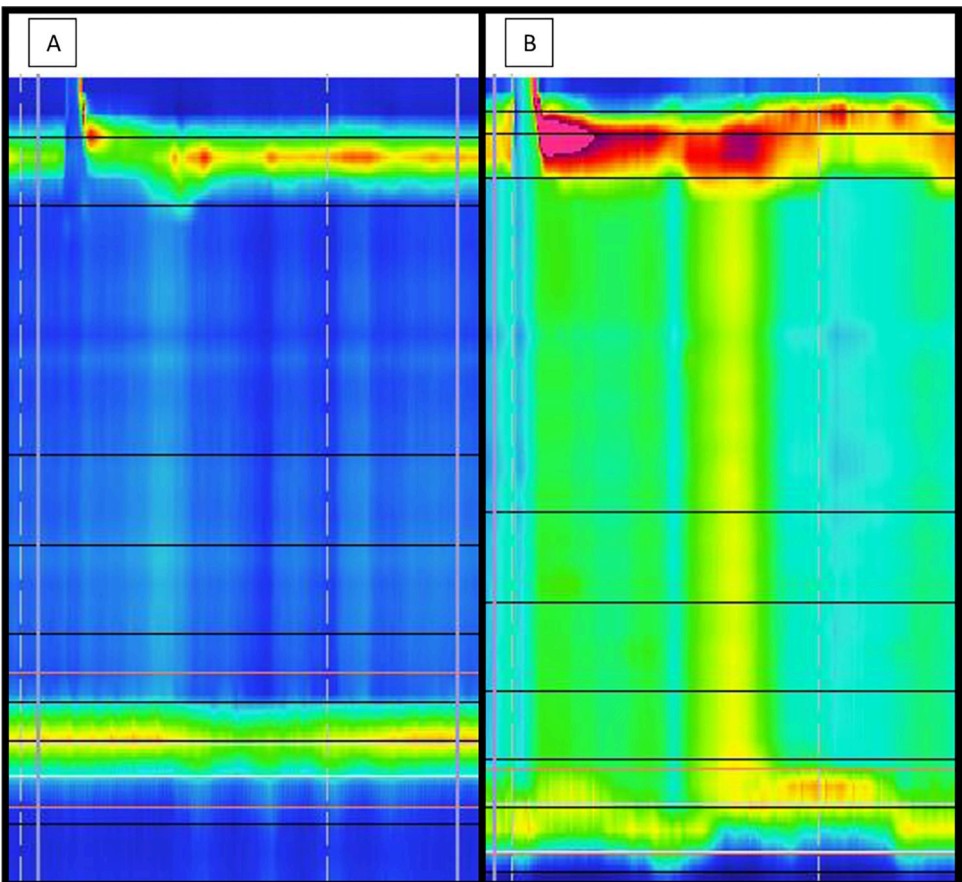

**Fig 1. Achalasia subtypes.** 1A: achalasia type 1: absent peristalsis and failure to relax LES. 1B: achalasia type 2 with panesophageal pressurization and failure to relax LES.

**Peroral endoscopic myotomy (POEM).** In 2009, an endoscopic means to perform the myotomy via a submucosal tunnel was developed; Peroral Endoscopic Myotomy (POEM) [16]. POEM integrates the theoretical advantages of both endoscopic dilation (no skin incisions, decreased pain, and less blood loss) and LHM (durable surgical myotomy and single procedure). To date, POEM has been increasingly used to treat adult achalasia patients due to its excellent outcomes and minimal risk of major adverse events [11, 17–19]. A large randomized controlled trial in adults comparing efficacy of POEM vs EBD, reported a one year success rate (Eckardt score ≤3) of 92.2% vs 70% respectively, in the absence of major complications for both procedures [20]. The 5 year success rate was 81% vs 40% for POEM vs EBD respectively, without intervention-related serious adverse events for both procedures. [21].

A recently published systematic review on outcomes of POEM in the pediatric population, evaluated results of 14 studies (n = 419; median age 14 years; 59% male). The studies available were mostly single center, uncontrolled and retrospective, but nevertheless showed promising results with success rates (Eckardt score of ≤3) of approximately 92.5% at an overall follow-up of 1 to 2 years [22]. Intention-to-treat analysis of four studies reporting success rates >2 years post-POEM (median of 26–40 months) showed a sustained success rate of 89%. To date, randomized controlled trials evaluating the two endoscopic therapies in children are lacking. Only one small pediatric study retrospectively compared the efficacy and safety of EBD and POEM (EBD = 9 and POEM = 12 patients) in terms of treatment success (Eckardt Score ≤ 3),

need for re-treatment and complications [23]. Treatment success of POEM was 100% at 12 and 36 months follow-up, whereas for EBD, treatment success was only achieved in 66.7% of patients at the respective time-points. None of the procedures was associated with serious complications. Two patients who were treated with POEM developed symptomatic esophagitis and were successfully treated with proton pump inhibitors (PPI) [23].

The aim of this study is to compare efficacy and safety of POEM vs. EBD as primary treatment for achalasia in children.

## Methods

### Study design

This is a prospective open international multicenter block-randomized and center-stratified controlled clinical trial in which POEM will be compared with EBD as first-step treatment for children with achalasia. Treatment outcome will be evaluated by a blinded end-point committee (PROBE design) [24].

**Patient and public involvement.** Patients were not involved in the design of this study.

**Sample size.** A recent retrospective review of 21 pediatric achalasia patients, of which 12 received POEM and 9 EBD, reported treatment success (Eckardt score ≤ 3) at 12 months after initial treatment to be 100% for POEM at all follow up moments and 66.7% for EBD patients respectively [17]. However, in this study no repeated EBD-treatment was allowed in case of clinical recurrence. In adults, 76% of patients treated according to our protocol, had clinical success after 12 months [10]. Age is known to be a factor in predicting success-rates of EBD with a slightly worse outcome in younger patients [3].

Taken all this together, we assumed a success rates of 72.5% for EBD and 92.5% for POEM in children after one year, we estimated that with 56 patients in each group, the study would have 80% power (based on a two group $\chi^2$ test) to detect a significant difference in success rate between EBD and POEM with a two-sided alpha level of 0.05. To cope with an estimated 10% loss to follow-up, we aim to enroll a total number of 126 patients (63 patients in each group). It is assumed that patient enrollment will be accomplished within 5 years, after approval from all local institutional review boards (IRBs).

**Recruitment procedure.** Treatment naïve pediatric patients diagnosed with achalasia type I or II who visit the outpatient clinic will be recruited by their treating physician. Due to the extremely low incidence of type III achalasia in pediatrics and the worse prognosis compared to type I and II, literature and expert-opinion based recommendations suggest to treat type III with POEM. Randomizing these patients to EBD with a high a priori risk of relapses is therefore considered unethical [25]. These patients were therefore excluded from this study.

*Inclusion criteria.* In order to be eligible to participate in this study, a subject must meet all of the following criteria:

- Eckardt score > 3 [26]

- presence of a HRM pattern consistent with achalasia type I or II according to the latest Chicago classification (CC) criteria

- Age 7 up to and including 17 years at the time of screening visit

  Exclusion criteria.

- Achalasia type III

- Previous surgical or endoscopic achalasia treatment

- Previous surgery of the upper gastrointestinal tract

- Known coagulopathy

- Known Liver cirrhosis and/or esophageal varices

- Known LA grade $\geq$B esophagitis

- Known Barrett's esophagus

- Known pregnancy at time of treatment

- Stricture of the esophagus

- Known presence of malignant or premalignant esophageal lesions

- Hiatal hernia > 1cm based on HRM measurement

- Extensive, tortuous dilation (>7cm luminal diameter, S shape) of the esophagus

- Barium esophagram suggestive of other pathologies

**Informed consent.** All patients and their parents/caregivers receive a written information letter and informed consent form. Randomization can only take place after patients (if $\geq$ 12 years) and all legal guardians (if patient < 16 years) signed the informed consent form.

**Ethical approval.** This study has been approved by the Medical Ethics Research Committee of the AMC hospital (NL68967.018.20).

**Trial registration.** This trial is registered in the ISRCTN registry. Study registration number: ISRCTN74448884. The trial was registered after enrolment of the first study participant because the process of registration took longer than expected. Only one patient was randomized before registration. The authors confirm that all ongoing and related trials for this intervention are registered.

**Randomization procedure, blinding and treatment allocation.** Randomization will be performed by using a web-based program (Castor electronic data capture (EDC) system) and will be stratified per center. To ensure balance of treatment groups with respect to experience in POEM and EBD per center, stratified block randomization with variable block size of 2, 4 or 6 in order to minimize selection bias will additionally be performed within each center.

Castor EDC will generate the allocation sequence. Allocation concealment is applied to all investigators, clinicians and patients.

**Treatment of subjects.** 'Initial therapy' is defined as:

- POEM: a single POEM procedure

- EBD: a series of two EBD sessions (30-35mm), scheduled 2–4 weeks apart.

- In case of *persisting* symptoms, as evaluated 7 days after the second EBD session, a third EBD session (35mm in children <12 years; 40mm in children $\geq$12 years old) will be performed 2–4 weeks after the second procedure. Persisting symptoms are defined as an Eckardt score>2. This definition differs from *recurrent symptoms* as the evaluation of weight loss within a week after treatment is considered inaccurate. Therefore a lower Eckardt score is accepted to reflect persisting symptoms.

- In case of *recurrent symptoms* $\geq$7 days after the second EBD session), (Eckardt score >3) and evidence of active disease on barium esophagram and/or HRM, a second series of two EBD 2–4 weeks apart will be performed as part of the initial therapy. For this third series a 30 & 35mm balloon is used in children <12 years and a 35 & 40mm balloon is used in children $\geq$12 years old).

- If a patient has already received an EBD because of persisting symptoms <7 days, persisting symptoms thereafter or recurrent symptoms are always considered a treatment failure.

In this trial, POEM and EBD will be performed by experienced (pediatric) endoscopists, who have independently performed over 15 procedures in children. Patients will be treated under general anesthetics.

**EBD procedure.** The first dilation is performed with a 30mm non-compliant balloon which is positioned at the esophagogastric junction (EGJ) over a guidewire and under fluoroscopic control. The EGJ is dilated with a pressure of 5 psi for 1 minute, followed by a pressure of 8 psi for 1 minute. Two to four weeks after the initial dilation, a second dilation will be performed. The procedure is identical to the first, but instead of 30mm, a 35mm balloon will be used.

If a third EBD session or second series (see above under initial therapy) is needed within the study period, this is considered part of the therapy. If, after this additional treatment session/series, symptoms recur again during the study period, this is considered treatment failure.

**POEM procedure.** The endoscopic procedure will be recorded on video to make evaluation afterwards possible. A forward-viewing upper endoscope is used with a transparent distal cap. Carbon dioxide gas is used for insufflation. An endoscopic dissection knife is used to access the submucosa, to create the submucosal tunnel, and also to divide circular muscle fibers over a minimum length of 6 cm in the esophagus, and 2 cm onto the cardia according to the standards of surgical myotomy. An electrogenerator is used with Endocut Q mode (effect 2) to open the mucosa, and spray coagulation mode (effect 2, 50 watt) to dissect the submucosa and divide the muscle fibers. A coagulating forceps is used for hemostasis if needed. Closure of the mucosal entry site is performed using standard endoscopic clips. Patients are kept nil per mouth until the next morning and are then offered water. If they can swallow the water without problems, they are discharged with a liquid diet (IDDSI 2) for a week and a pureed diet for another week (IDDSI 4). After two weeks they are allowed a normal diet.

**Retreatment in case of investigational treatment failure.** The need for retreatment (i.e. presence of treatment failure) is evaluated by a central committee blinded to the treatment arm. This committee takes into account:

- Symptom persistence or recurrence (despite a third EBD session or second series after initial EBD treatment, see page 8): Eckardt score > 3 (27) AND

- Barium esophagram with esophageal stasis at T = 1 minute, and/or HRM (performed if barium esophagram is inconclusive) with IRP4 above upper limits of normal [4, 27].

The type of retreatment (LHM, POEM, EBD or intra-sphincteric botox injection) will be at the discretion of the attending physician's team.

## Outcome measures

**Main study parameter/endpoint.** Primary outcome measure is the need for retreatment (*see 'Retreatment in case of initial treatment failure'*). Primary outcome will be evaluated at 12 months follow-up.

## Secondary study parameters/endpoints

- Achalasia symptoms (Eckardt score); Health-related- and disease specific QoL (Pediatric Quality of Life inventory (PedsQL) 4.0, PedsQL-GI, and Achalasia disease specific QoL questionnaire (DSQoL)) [28, 29]

**Table 1. Study parameters of HRM, pH-MII and EGD.**

| Measurement | Outcomes |
|---|---|
| **High resolution manometry** | Achalasia subtype I / II according to CC (17) |
| | LES resting pressure (mmHg) |
| | LES integrative relaxation pressure (mmHg) |
| | Distal contractile integral (mmHg·s·cm) |
| | Distal latency (S) |
| **pH-impedance** | Acid exposure time (% pH < 4)<br>• total<br>• upright position<br>• supine position |
| | Number of reflux episodes<br>• upright vs supine<br>• acid (pH < 4)/ weakly acidic (pH 4–7) / alkalic (pH >7) |
| | Symptom indices<br>• Symptom Index<br>• Symptom sensitivity index<br>• Symptom association probability |
| | Esophageal stasis defined by bolus presence time |
| **Esophagogastroduodenoscopy** | • macroscopic abnormalities<br>• LA esophagitis classification<br>Additional parameters at T = 12 months:<br>• microscopic signs of inflammation<br>• microscopic count of eosinophils |

To minimize the risk of bias, all EGD; pH-MII; HRM and contrast esophagram will be analyzed by a blinded committee.

- Reflux symptoms (reflux disease questionnaire, RDQ) [30]

- LA grade (EGD)

- 24 hour pH-impedance (pH-MII) measurement parameters (Table 1)

- HRM parameters (Table 1)

- Stasis on contrast esophagram 1 minute after ingestion of barium

- Procedure times

- Complications (any unwanted events that arise following treatment and/or that are secondary to the treatment)

- Severe: resulting in admission >24 hours or prolongation of an already planned admission of >24 hours, admission to a medium or intensive care unit, additional endoscopic procedures, blood transfusion or death.

- Mild: all other complications.

  **Follow-up scheme for participants.** *Baseline, 1 week post-initial therapy, 3 weeks post-initial therapy, 3 months, 6 months and 12 months follow-up (*Figs 2 and 3*)*

- Standard achalasia care:

- Body weight and height and body mass index (BMI) including respective *z*-scores

- Medical history including Eckardt Score [26]

- PedsQL 4.0 [28, 31]

SPIRIT checklist.*

| | Enrolment | | STUDY PERIOD | | | | | | | | | | | | Close-out |
|---|---|---|---|---|---|---|---|---|---|---|---|---|---|---|---|
| | | | Post-allocation | | | | | | | | | | | | |
| TIMEPOINT** | $-t_1$ | $t_0$ | $t_1$ | $t_2$ | $t_3$ | $t_4$ | $t_5$ | $t_6$ | $t_7$ | $t_8$ | $t_9$ | $t_{10}$ | $t_{11}$ | $t_{12}$ |
| **ENROLMENT:** | | | | | | | | | | | | | | |
| **Eligibility screen** | X | | | | | | | | | | | | | |
| **Informed consent** | X | | | | | | | | | | | | | |
| **Allocation** | | X | | | | | | | | | | | | |
| **INTERVENTIONS:** | | | | | | | | | | | | | | |
| **POEM or EBD** | | X | | | | | | | | | | | | |
| **ASSESSMENTS:** | | | | | | | | | | | | | | |
| **Eckardt score** | X | | ←————————————————————————→ | | | | | | | | | | | |
| **Quality of life questionnaires** | X | | | | X | | X | | | | | | | X |
| **HRM** | X | | | | | | | | | | | | | X |
| **Barium Swallow** | X | | | | | | | | | | | | | X |
| **Gastroscopy** | | | | | | | | | | | | | | X |

*Recommended content can be displayed using various schematic formats. See SPIRIT 2013 Explanation and Elaboration for examples from protocols.
**List specific timepoints in this row.

**Fig 2. SPIRIT checklist.** *Recommended content can be displayed using various schematic formats. See SPIRIT 2013 Explanation and Elaboration for examples from protocols.**List specific timepoints in this row.

- PedsQL GI [32]

- Additional study measurements and questionnaires

- Disease specific QoL [29]

- Reflux disease questionnaire [30]

Additional tests performed at 12 months follow-up:

- Barium esophagram

- HRM

- pH-MII

- EGD

**Risks and benefits.** If a pediatric patient is diagnosed with achalasia, invasive treatment is always needed. Because EBD and POEM are currently both used as therapeutic options for

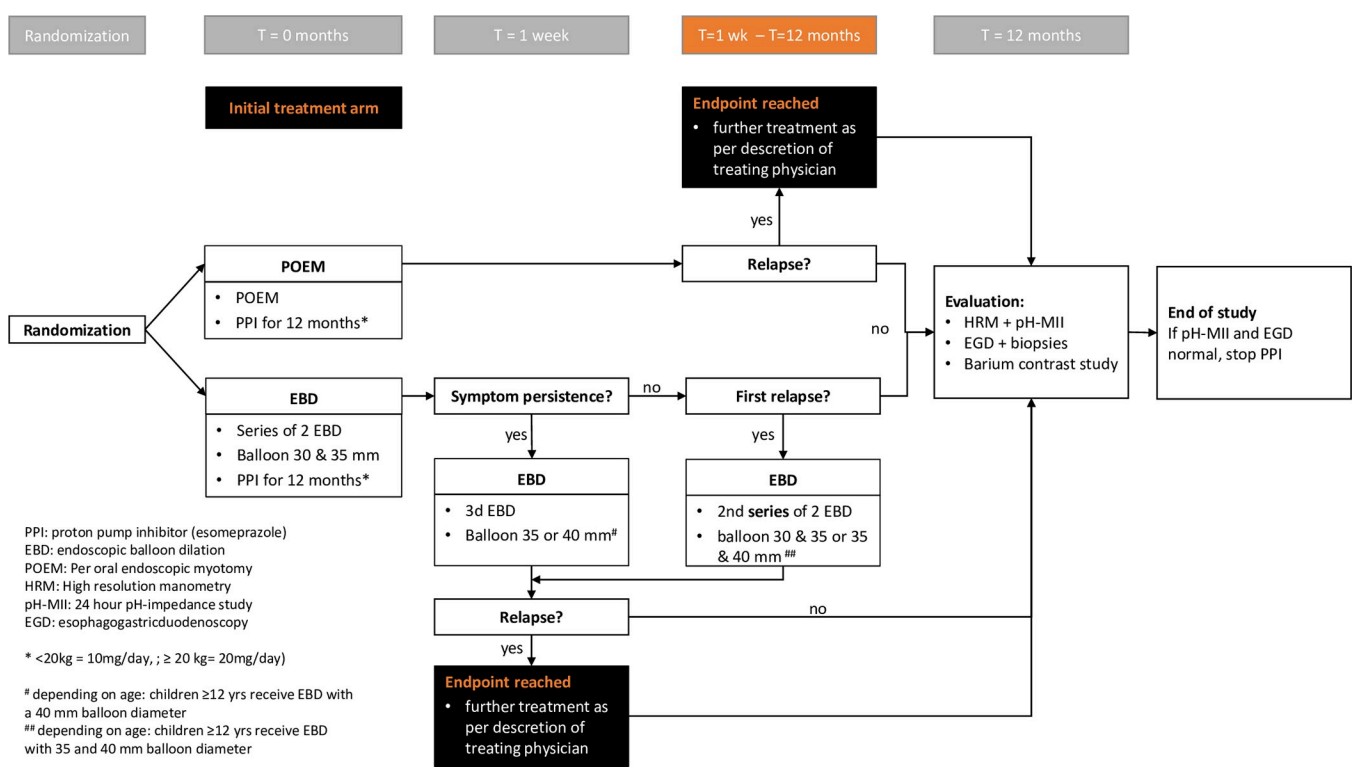

**Fig 3. Study flowchart.** PPI: proton pump inhibitor (esomeprazole) EBD: endoscopic balloon dilation POEM: Per oral endoscopic myotomy HRM: High resolution manometry pH-MII: 24 hour pH-impedance study EGD: esophagogastricduodenoscopy. *<20kg = 10mg/day; ≥20 kg = 20mg/day. # depending on age: children ≥12 yrs receive EBD with a 40 mm balloon diameter ## depending on age: children ≥12 yrs receive EBD with 35 and 40 mm balloon diameter.

children with achalasia, children will not have any additional risk when participating in this trial.

**Risk and complications of EBD.** Complications due to EBD are uncommon and include perforation (data from retrospective pediatric studies: n = 5/209 (2.4%) and 1/60 (1.7%) respectively [28]), post-procedural pain (n = 1/10,10%) and fever (n = 2/10,20%). Therapy for perforations is mostly conservative with food and drink restriction and intravenous antibiotic therapy. In a randomized controlled trial in adults comparing efficacy of POEM vs EBD, two serious adverse events occurred after EBD (n = 1/66 (1.5%) perforation; n = 1/66 (1.5%) post-procedural pain) [20].

**Risk and complications of POEM.** POEM has been shown feasible and safe in children [11, 12, 17]. In a randomized controlled trial in adults comparing efficacy of POEM vs EBD, no serious adverse events occurred after POEM [20].

Serious adverse events related to the endoscopic procedure that could theoretically occur during the procedure are bleeding or esophageal perforation. Treatment of these complications can instantaneously be performed during the procedure by clipping the bleeding vessel or perforation. In exceptional cases a surgical procedure could be needed.

Additionally, complications related to $CO_2$ insufflation during the procedure may occur. These include retroperitoneal $CO_2$, capnoperitoneum or a pneumoperitoneum. A pneumoperitoneum can be relieved by a puncture during the POEM procedure. Postoperative retroperitoneal $CO_2$ is usually self-limiting and additional treatment is not needed. All the additional measurements that are performed before and after treatment are safe procedures and routinely performed in the clinical setting.

**Reflux surveillance.** GERD may develop post-achalasia intervention due to disruption of the LES. A recently performed randomized controlled trial in adults with achalasia showed a higher incidence of GERD in patients who were treated with POEM (31% grade A; 4% grade B, 8% grade C) compared to those treated with EBD (7% grade A), p = 0.002. In a recently published literature review, GERD was present in 8% of children treated with EBD vs 28% of children treated with POEM [5]. However, in these pediatric studies, GERD was defined with varying criteria and diagnostic tests and many studies were retrospective in design. A true estimate of the occurrence of GERD post-achalasia treatment, will only be possible by prospective evaluation using standardized tools (i.e. EGD with biopsies and pH-MII). As a precautionary measure, all study patients will take PPIs orally in a single daily dose starting at the day after treatment (esomeprazole, <20kg = 10mg/day,; $\geq$ 20 kg = 20mg/day for at least one year (+/- 6 weeks) after initial therapy. At 12 months after randomization, an EGD, HRM and pH-MII will be performed after cessation of PPI 4–6 weeks prior to investigations. In case of abnormal pH-MII and/or EGD results, the attending physician will follow-up and manage GERD according to latest recommendation guidelines of pediatric GERD [33]. In case of normal pH-MII and EGD results, PPIs can be ceased.

**Adverse events.** Adverse events are defined as any undesirable experience occurring to a subject during the study, whether or not considered related to the intervention. All adverse events reported spontaneously by the subject or observed by the investigator or his staff will be recorded.

**Data Safety Monitoring Board (DSMB).** An independent DSMB consisting of an epidemiologist, a pediatric gastroenterologist and an adult gastroenterologist will evaluate safety of participants after 50 participants have reached 1 year follow up.

**Statistical analysis.** All normally distributed data will be presented as means and standard deviations (SD). Data where normal distribution cannot be assumed, will be reported as median and range. Categorical data will be summarized using frequencies and corresponding percentages. The success rate of the two procedures and their odds ratio will be analyzed using logistic regression with adjustment for stratification factors. Superiority of POEM is shown when the primary outcome (need for retreatment) is significantly lower in the POEM treatment arm (defined as p<0.05). Data will be shown in point estimates and 95% confidence intervals. Comparisons between groups will be made using multiple linear regression analysis for continuous data, if necessary after (log)transformation and multiple logistic regression analysis for categorical data for secondary outcomes, adjusting for stratification factors. Analyses will be performed according to an intention-to-treat protocol.

**Dissemination.** The results of this RCT will be presented at national and international conferences and published in an international medical journal.

## Discussion

Achalasia is a life-long esophageal disorder, in which children experience significant morbidities associated with this condition, both pre- and post-therapy [1, 34]. Recurrent symptoms, frequent hospital visits and relapse therapies impact quality of life in these patients [1, 35]. It is therefore important to evaluate how children should best be treated to achieve long-term treatment success without development of procedure-related complications. Although studies in adult achalasia patients are available, it is known that younger patients have worse clinical success rates of EBD [31].Therefore, the results of adult studies do not necessarily reflect therapeutic successes in the pediatric population.

To date, clinical management of pediatric achalasia is based on small and often retrospective pediatric studies. The lack of prospective randomized trials may be caused by the rarity of

the disease which comes with difficulties to set-up and perform a well-established trial [36]. Studies in rare diseases always require multiple study-sites to obtain a powerful sample size. The setup of various study-sites is challenging due to differences between regulation amongst centers and countries. Additionally, studies in rare diseases like achalasia are often initiated by academic centers of expertise, without involvement of pharmaceutical companies. Although the independency of such trials can be seen as a study strength, it also comes with the lack of financial support. Study results of this trial will improve the care for pediatric achalasia patients and may improve quality of life and possibly reduce number of relapse treatments.

## Supporting information

**S1 Checklist. SPIRIT 2013 checklist: Recommended items to address in a clinical trial protocol and related documents\*.**
(DOCX)

**S1 File.**
(PDF)

**S2 File.**
(PDF)

## Author Contributions

**Conceptualization:** Carlijn Mussies, Marinde van Lennep, Johanna H. van der Lee, Marc. A. Benninga, Barbara A. Bastiaansen, Paul Fockens, Albert J. Bredenoord, Michiel P. van Wijk.

**Funding acquisition:** Michiel P. van Wijk.

**Methodology:** Carlijn Mussies, Marinde van Lennep, Johanna H. van der Lee, Michiel P. van Wijk.

**Project administration:** Michiel P. van Wijk.

**Resources:** Michiel P. van Wijk.

**Supervision:** Marc. A. Benninga, Albert J. Bredenoord, Michiel P. van Wijk.

**Writing – original draft:** Carlijn Mussies, Marinde van Lennep, Maartje J. Singendonk, Michiel P. van Wijk.

**Writing – review & editing:** Carlijn Mussies, Marinde van Lennep, Johanna H. van der Lee, Marc. A. Benninga, Barbara A. Bastiaansen, Paul Fockens, Albert J. Bredenoord, Michiel P. van Wijk.

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
