## [Decision Letter · Decision Letter 0]

6 Mar 2023

PONE-D-22-31539Protocol for an international multicenter randomized controlled trial assessing treatment success and safety of peroral endoscopic myotomy vs endoscopic balloon dilation for the treatment of achalasia in childrenPLOS ONE

Dear Dr. van Wijk,

Thank you for submitting your manuscript to PLOS ONE. After careful consideration, we feel that it has merit but does not fully meet PLOS ONE’s publication criteria as it currently stands. Therefore, we invite you to submit a revised version of the manuscript that addresses the points raised during the review process.

We look forward to receiving your revised manuscript.

Kind regards,

Antonio Brillantino

Academic Editor

PLOS ONE

2. Thank you for submitting your clinical trial to PLOS ONE and for providing the name of the registry and the registration number. The information in the registry entry suggests that your trial was registered after patient recruitment began. PLOS ONE strongly encourages authors to register all trials before recruiting the first participant in a study. As per the journal’s editorial policy, please include in the Methods section of your paper: 1) your reasons for your delay in registering this study (after enrolment of participants started); 2) confirmation that all related trials are registered by stating: “The authors confirm that all ongoing and related trials for this drug/intervention are registered.

“For Wishdom Foundation

https://www.forwishdom.org/”

4. Thank you for stating the following in the Acknowledgments/ Funding Section of your manuscript:

“This study is funded by for Wishdom.”

“For Wishdom Foundation

https://www.forwishdom.org/”

6. Please include a separate caption for each figure in your manuscript.

7. Please include a copy of Table 1 which you refer to in your text on page 11.

Additional Editor Comments:

This study describes the protocol of a randomized controlled trial comparing endoscopic myotomy with ballon dilation in the treatment of esophageal achalasia in children. Overall, the outcomes of the study are clearly exposed and the statistical analysis adequately formulated. According with the reviewer’s indications, the study need some minor revisions and overall, is suitable for publication after minor changes.

Reviewers' comments:

Reviewer's Responses to Questions

**Comments to the Author**

1. Does the manuscript provide a valid rationale for the proposed study, with clearly identified and justified research questions?

Reviewer #1: Yes

2. Is the protocol technically sound and planned in a manner that will lead to a meaningful outcome and allow testing the stated hypotheses?

Reviewer #1: Yes

3. Is the methodology feasible and described in sufficient detail to allow the work to be replicable?

Reviewer #1: Yes

4. Have the authors described where all data underlying the findings will be made available when the study is complete?

Reviewer #1: Yes

5. Is the manuscript presented in an intelligible fashion and written in standard English?

Reviewer #1: Yes

6. Review Comments to the Author

You may also provide optional suggestions and comments to authors that they might find helpful in planning their study.

Reviewer #1: In this study protocol, a multi-center two-arm randomized control trial is being proposed to compare the efficacy of and safety of Peroral Endoscopic Myotomy to endoscopic balloon dilation as treatments for achalasia in children. The primary outcome will be need for retreatment due to treatment failure and will be evaluated at the 12-month follow-up time.

Minor revisions:

1- Line 173: Indicate the statistical testing method which achieves 80% power.

2- Lines 341-6: Provide corresponding percentages for the fractions shown.

3- Line 386 Statistical analysis: Indicate how categorical variables will be summarized. Typically frequencies and corresponding percentages are provided.

4- State how adverse event data will be summarized. Indicate if a standard method for collecting adverse events will be implemented such as the Common Terminology Criteria for Adverse Events (CTCAE) version 5.

7. PLOS authors have the option to publish the peer review history of their article (what does this mean?). If published, this will include your full peer review and any attached files.

Reviewer #1: No

---

## [Author Response · Author response to Decision Letter 0]

21 Mar 2023

Dear editor, 

The manuscript is amended to meet all journal requirements and all minor revisions are implemented. 

Kind regards

---

## [Decision Letter · Decision Letter 1]

25 May 2023

Protocol for an international multicenter randomized controlled trial assessing treatment success and safety of peroral endoscopic myotomy vs endoscopic balloon dilation for the treatment of achalasia in children

PONE-D-22-31539R1

Dear Dr. van Wijk,

We’re pleased to inform you that your manuscript has been judged scientifically suitable for publication and will be formally accepted for publication once it meets all outstanding technical requirements.

Kind regards,

Antonio Brillantino

Academic Editor

PLOS ONE

Additional Editor Comments (optional):

Reviewers' comments:

Reviewer's Responses to Questions

**Comments to the Author**

1. Does the manuscript provide a valid rationale for the proposed study, with clearly identified and justified research questions?

Reviewer #2: Yes

Reviewer #3: No

2. Is the protocol technically sound and planned in a manner that will lead to a meaningful outcome and allow testing the stated hypotheses?

Reviewer #2: Yes

Reviewer #3: Partly

3. Is the methodology feasible and described in sufficient detail to allow the work to be replicable?

Reviewer #2: Yes

Reviewer #3: Yes

4. Have the authors described where all data underlying the findings will be made available when the study is complete?

Reviewer #2: Yes

Reviewer #3: No

5. Is the manuscript presented in an intelligible fashion and written in standard English?

Reviewer #2: Yes

Reviewer #3: No

6. Review Comments to the Author

You may also provide optional suggestions and comments to authors that they might find helpful in planning their study.

Reviewer #2: Thank you for giving me the valuable opportunity to review this manuscript.

The author is planning a trial to determine whether POEM or EBD is more appropriate as initial treatment in children with achalasia. The results that would be obtained from this study are clinically very important and the methodology is appropriate.

Some questions are noted below.

1. Other papers on pediatric achalasia patients have reported invasive treatment, including POEM, for patients under 7 years of age. It would be better to provide a rationale for the lower age limit of 7 years in this study.

2. Some patients with achalasia may not be getting enough nutrition and may not be growing well enough. Is there a need to set a lower weight limit?

Reviewer #3: This is a multicentered randomized controlled trial comparing POEM AND pneumatic dilation for pediatric achalasia. There are several concerned aspects:

1.I am not sure whether a protocol without any trial results could be publised. There was no result addressing the efficacy or safty of POEM or dilation.

2.In page 4, line 88, “esophageal spasm” is present in type 3 achalasia.

3.In Introduction, mediacation such as nitrites is missing. Medication is an optional therapy.

4.In page 6,line 158. “Patients were not involved” in the study design. But in line 160, “21 pediatric patients” were involved. This is weired. Please clarify this.

5. In page 13, line 363, peumothorax is more frequent than pneumoperitium in POEM. Please check this.

6.In page 14, line 368, it was not necessary in discussing GERD in Methods.

7.Threre was no results in the article.

8.The discussion is too simple to draw a conclusion.

7. PLOS authors have the option to publish the peer review history of their article (what does this mean?). If published, this will include your full peer review and any attached files.

Reviewer #2: **Yes: **Shinwa Tanaka

Reviewer #3: No

---

## [Editor Report · Acceptance letter]

5 Jun 2023

PONE-D-22-31539R1 

Protocol for an international multicenter randomized controlled trial assessing treatment success and safety of peroral endoscopic myotomy vs endoscopic balloon dilation for the treatment of achalasia in children 

Dear Dr. van Wijk:

I'm pleased to inform you that your manuscript has been deemed suitable for publication in PLOS ONE. Congratulations! Your manuscript is now with our production department. 

Kind regards, 

on behalf of

Dr Antonio Brillantino 

Academic Editor

PLOS ONE